# Learning Unseen Modality Interaction

**Yunhua Zhang, Hazel Doughty, Cees G.M. Snoek**
University of Amsterdam

## Abstract

Multimodal learning assumes all modality combinations of interest are available during training to learn cross-modal correspondences. In this paper, we challenge this modality-complete assumption for multimodal learning and instead strive for generalization to unseen modality combinations during inference. We pose the problem of *unseen modality interaction* and introduce a first solution. It exploits a module that projects the multidimensional features of different modalities into a common space with rich information preserved. This allows the information to be accumulated with a simple summation operation across available modalities. To reduce overfitting to less discriminative modality combinations during training, we further improve the model learning with pseudo-supervision indicating the reliability of a modality's prediction. We demonstrate that our approach is effective for diverse tasks and modalities by evaluating it for multimodal video classification, robot state regression, and multimedia retrieval. Project website: https://xiaobai1217.github.io/Unseen-Modality-Interaction/.

## 1   Introduction

Real-world machine learning models must operate on a wide variety of platforms ranging from autonomous vehicles to wearable devices. This requires effective combination of data from the diverse suite of sensors available. Multimodal learning, which learns how to combine multiple different data sources to make a prediction, has therefore become a central problem in machine learning [3]. Learning correspondences between modalities is a challenging problem as modalities differ in their input representations, learning dynamics, and discriminability for a target task. Recent multimodal learning methods [16, 28, 36, 32, 22] thus require the training data to have all modalities available for every sample. We call this modality-complete learning. However, this constraint will not always hold in the real world as it may be infeasible to collect sufficient training data which jointly measures data across all sensors. Similarly, sensors could be added or removed at any time. It is thus important to make multimodal learning methods able to handle the mismatch between modality combinations at training and inference, which is the topic of this paper.

We propose the multimodal learning problem that is able to infer predictions over modality combinations unseen during training. Several previous works [45, 23, 26, 32, 22] have increased robustness to the data with missing modalities, *i.e.*, modality-incomplete data. For example, Miech *et al.* [26] predict a weighting to combine unimodal predictions, while Shvetsova *et al.* [32] utilize cross-modal attention for multiple modality combinations to obtain the prediction. Albeit successful, these methods still require a portion of the data to have all modalities of interest to learn the relative importance of different modalities [26] or the cross-modal correspondences [32, 22, 45, 23]. Thus, they cannot handle unseen modality combinations at inference. Different from previous works, we propose to relieve the need for modality-complete training data and generalize to unseen modality combinations.

In this paper, our main contribution is to establish the problem of unseen modality interaction, where we strive to learn a multimodal model from only modality-incomplete data that can handle unseen modality combinations at test time. The challenge is to effectively combine modalities without any training data of the modality combinations that will be seen in inference. We propose to address this problem by effectively collecting information from available modalities while not explicitly

37th Conference on Neural Information Processing Systems (NeurIPS 2023).

learning cross-modal correspondences. Specifically, we project multidimensional features of different modalities into a common space, while keeping as much distinguishing information as possible. To reduce overfitting to unreliable modality combinations during training, we further improve the model learning by pseudo-supervision with dual branch prediction. To evaluate this new problem, we reorganize existing datasets and tasks for video classification, robotic state regression, and multimedia retrieval on a variety of modality combinations. Our approach enables effective unseen modality interaction and is more robust to scenarios where some modalities are corrupted by severe noise.

## 2   Related Work

**Modality-Complete Learning**. The field of multimodal learning is extensive, e.g., [16, 14, 20, 32, 44, 5, 36, 34, 38, 31, 33, 2, 30], as different modalities can provide complementary information. A simple approach is to integrate information across modalities inputting concatenated unimodal features to linear layers [16, 44, 5]. Several works [14, 20, 34, 38, 31, 2, 30] instead capture the correlations between different modalities with cross-modal attention, enhancing the discriminative features for each modality. Some recent works [9, 28, 32, 36] utilize self-attention to gain both cross-modal and intra-modal attention. For instance, Gabeur *et al*. [9] directly use the features of different modalities as the inputs to a transformer. Nagrani *et al*. [28] make this more efficient by condensing the relevant information per modality with attention bottlenecks. Wang *et al*. [36] instead dynamically replace uninformative feature tokens among transformer layers with inter-modal features. Another line of research learns multimodal representations by utilizing the correspondences in multimodal data, e.g., [43, 33]. Zhang *et al*. [43] use audio to adapt their visual model to distribution shift, while Swetha *et al*. [33] learn projection functions to transform unimodal features into a joint embedding space with the semantic structures preserved. Although these works achieve impressive performance, they assume all data in both training and testing have the same modalities. We instead learn a model suitable for unseen modality interactions at test-time, which are not present in the training data.

**Modality-Incomplete Learning**. Some prior works [23, 42, 45, 22, 26, 32, 37] do not assume all data to be modality-complete and are more robust to missing modalities at test-time [45, 22, 42, 29, 37] or in training [23, 26, 32]. To obtain test-time predictions with a single modality, Recasens *et al*. [29] apply masks to the attention in the multimodal transformer. Ma *et al*. [22] improve robustness to missing modalities during testing by searching for the optimal multimodal learning strategies. Miech *et al*. [26] instead learn from modality-incomplete training data, by predicting a weighting to combine unimodal predictions. Alternatively, Ma *et al*. [23], Zhao *et al*. [45] and Wang *et al*. [37] hallucinate the missing modalities according to the available ones. Shvetsova *et al*. [32] leave out the positional encodings and use a combinatorial loss so that the transformer can accept any modality combination. While these works are robust to data with missing modalities, they still require some training data to be modality-complete. We remove this assumption and learn a model able to make predictions at test-time for modality combinations not seen during training.

**Modality-Agnostic Models**. While the multimodal learning methods discussed above make use of unimodal encoders, several works [1, 15, 11] study modality-agnostic models, which possess the flexibility to take any modality as input. For instance, Jaegle *et al*. [15] propose the Perceiver model that can handle various modalities, by leveraging an asymmetric attention mechanism to distill inputs into a tight latent bottleneck. Akbari *et al*. [1] present a way for video, audio, and text modalities to share the same transformer model when trained by a self-supervised contrastive objective. Girdhar *et al*. [11] develop a modality-agnostic vision model trained by unpaired image, video and single-view 3D data as various visual modalities have a lot in common (e.g., shapes). Even though these works achieve state-of-the-art results on various tasks, they only extract unimodal representations [15, 1, 11]. We instead learn multimodal representations from modality-incomplete data.

**Federated Multimodal Learning**. Similar to our task, federated learning [17, 25, 4, 40, 19] also aims to learn from unpaired data, collected from different devices. Specifically, the goal is to collaboratively train a model with data from multiple devices without centralizing the data. Commonly this is solved by training models on local devices separately and aggregating these models with the server, which creates an updated global model. Most relevant to our work, is the work by Yang *et al*. [41]. This goes a step further and handles local devices with different sensors, although it assumes some modality-complete data are available for model learning. Our task has a similar inspiration in that data may be collected from different devices. However, we assume that all training data are centrally available, but are modality-incomplete and we focus on learning for unseen modality interactions.

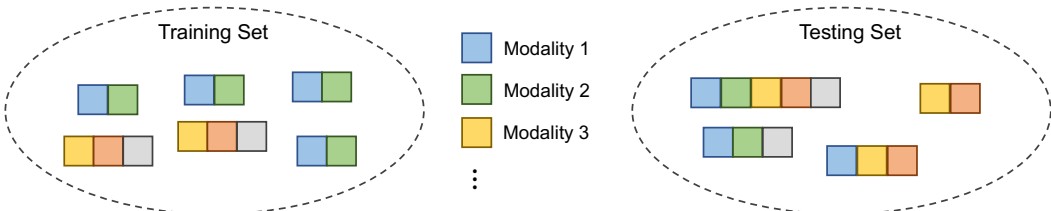

Figure 1: **Learning Unseen Modality Interactions**. The connected squares represent a sample, with each color indicating a different modality. Our goal is to learn from a modality-incomplete training set to make predictions for unseen modality combinations during inference. The modality combinations at inference could be either a union of or subsets of the modalities used in training.

## 3 Problem Definition

Our goal is to learn from a modality-incomplete training set, where some modality combinations do not have data available, to make predictions on unseen modality combinations during inference (as depicted in Figure 1). We denote a set of $n$ samples with $k$ modalities $\mathcal{M}=\{m_1, m_2, \cdots, m_k\}$ by $\mathcal{X}_{\mathcal{M}}=\{x_{\mathcal{M}}^1, \cdots, x_{\mathcal{M}}^n\}$. We aim to make predictions for a set of samples $\mathcal{X}_{\mathcal{M}^*}$ with available modalities $\mathcal{M}^*$, without having access to training data with the same complete set of modalities. Instead, the task is to learn from multiple sets of samples containing subsets of the modalities $\mathcal{M}^*$. For ease of explanation, we consider the scenario where we learn from two sets of samples $\mathcal{X}_{\mathcal{M}_1}$ and $\mathcal{X}_{\mathcal{M}_2}$ with different sets of modalities, *i.e.*, $\mathcal{M}_1 \neq \mathcal{M}_2$. However, this can easily be extended to more than two modality sets. Each modality set in training is a subset of the modalities seen at inference, *i.e.*, $\mathcal{M}_1, \mathcal{M}_2 \subset \mathcal{M}^*$, and together have all available modalities, *i.e.*, $\mathcal{M}^*=\mathcal{M}_1 \cup \mathcal{M}_2$. The objective is to make predictions using the available modalities during inference even though this modality combination never appears during model learning. Two main challenges exist. The first is integrating modality-specific features from different feature spaces without any modality-complete training data containing all modalities of interest. The second is reducing the overfitting to the specific modality combinations from the modality-incomplete training data.

## 4 Method

The inputs to our model are the raw signals of available modalities for a sample, from $\mathcal{X}_{\mathcal{M}_1} \cup \mathcal{X}_{\mathcal{M}_2}$ during training and $\mathcal{X}_{\mathcal{M}^*}$ at test time. After training, our model is able to integrate information from different modalities for prediction even though the modality combinations are unseen in training.

Figure 2 shows our method overview. Each modality $m$ of a sample is first encoded by a unimodal encoder to generate the unimodal features $\mathbf{F}_m$. These features from different modalities will be in different spaces with different dimensions. To tackle the challenge of integrating different modality combinations, we propose to project features of available modalities into a common space by our feature projection, while keeping as much distinguishing information as possible. To reduce overfitting to the modality combinations in training, we improve the supervision by introducing pseudo-labels, which indicate the reliability of different modalities, to learn our dual branch prediction.

**Feature Projection**. We use pre-trained unimodal encoders $\mathcal{E}_m(\cdot)$ to extract the features of each modality. These features are then flattened to tokens $\mathbf{F}_m=[\mathbf{f}_1, \cdots, \mathbf{f}_{k_m}], \mathbf{F}_m \in \mathbb{R}^{k_m \times d_m}$ with $k_m$ tokens and $d_m$ feature dimension for modality $m$. The feature tokens of different modalities are unaligned and in different feature spaces with different lengths $k_m$ and dimensions $d_m$. For example, audio tokens correspond to a specific range of time and frequency, while video tokens represent a specific spatial-temporal video segment. Existing works with modality-complete training data concatenate features of different modalities as input to a multimodal transformer. However, without modality-complete data in training, a multimodal transformer cannot learn to find cross-modal correspondences. Thus, we need a model that can handle a flexible combination of modalities to effectively learn from multiple sets of modality-incomplete data $\mathcal{X}_{\mathcal{M}_1}$ and $\mathcal{X}_{\mathcal{M}_2}$ and make predictions for modality-complete data $\mathcal{X}_{\mathcal{M}^*}$. Instead of concatenation we use the summation operation, which provides a natural way to accumulate information from any number of available modalities, as long as the modality-specific features are in a common feature space. However, projecting modality-specific features into a single vector in the common space as in previous works [24, 27] loses a lot of

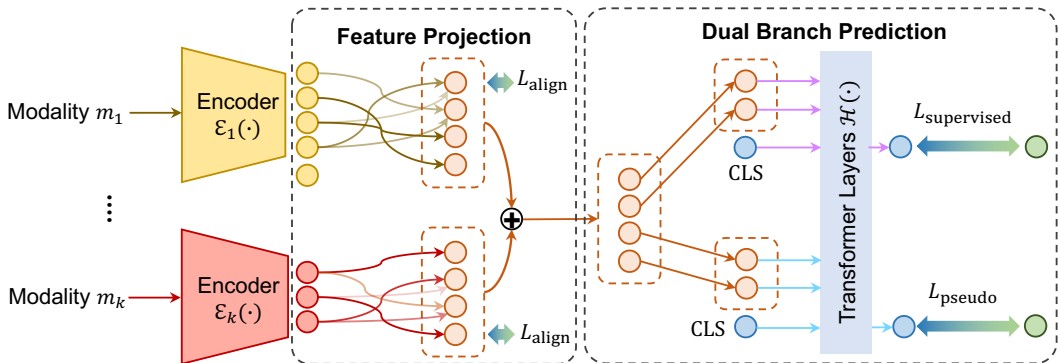

Figure 2: **Method Overview.** Given a sample collected from multiple sensors, each modality is first encoded by a unimodal encoder. To generalize to unseen modality combinations, our feature projection projects the unimodal features into a common space while keeping as much differentiating information as possible. We encourage feature alignment by applying $L_{\text{align}}$ to each modality in training. Then, we can obtain a multimodal representation by adding all modalities available for the input sample together. To reduce overfitting to the modality combinations in training, we propose our dual branch prediction. Features are first divided into two groups corresponding to the two branches. One branch is supervised by groundtruth labels via $L_{\text{supervised}}$ and the other by pseudo-labels via $L_{\text{pseudo}}$. During inference, the final prediction is the average of the two branches.

information, thus it cannot enable effective model learning. We instead introduce a feature projection module for projecting multidimensional features into a common space. This approach allows the projected features of different modalities to have the same length and dimension with each feature element having the same semantic meaning over multiple modalities.

Our feature projection module takes the unimodal features $\mathbf{F}_m = [\mathbf{f}_1, \cdots, \mathbf{f}_{k_m}], \mathbf{F}_m \in \mathbb{R}^{k_m \times d_m}$ as inputs and outputs the features in the common space $\hat{\mathbf{F}}_m = \mathbf{F}_m^{\text{T}} \mathbf{O}_m, \hat{\mathbf{F}}_m \in \mathbb{R}^{k^* \times d^*}$, where $k^*$ and $d^*$ indicate the length of the feature tokens and their dimension which are shared across modalities. To this end, we first project different unimodal features of different lengths $k_m$ to a fixed length $k^*$ via an attention matrix $\mathbf{O}_m = [\mathbf{o}_1, \cdots, \mathbf{o}_{k_m}], \mathbf{O}_m \in \mathbb{R}^{k_m \times k^*}$, which is obtained from $\mathbf{F}_m$ through several transformer layers. The matrix $\mathbf{O}_m$ is normalized using the Softmax operation so that $\sum_{j=1,\cdots,k_m} o_{k_m,j} = 1$. $\mathbf{O}_m$ indicates how to combine the input feature tokens to obtain each output feature token. Then, we obtain the reorganized features by $\mathbf{F}'_m = \mathbf{F}_m^{\text{T}} \mathbf{O}_m, \mathbf{F}'_m \in \mathbb{R}^{k^* \times d_m}$, and each feature token of $\mathbf{F}'_m$ is a combination of input tokens from $\mathbf{F}_m$. To project the reorganized features $\mathbf{F}'_m$ into the common space with the dimension $d^*$, we pass them through a fully connected layer to get $\hat{\mathbf{F}}_m = [\hat{\mathbf{f}}_m^1, \cdots, \hat{\mathbf{f}}_m^{k^*}], \hat{\mathbf{F}}_m \in \mathbb{R}^{k^* \times d^*}$. While each feature token in the resulting $\hat{\mathbf{F}}_m$ has the same semantic meaning across available modalities, we can add them together to obtain a multimodal representation $\hat{\mathbf{F}} = \sum_m \hat{\mathbf{F}}_m, \hat{\mathbf{F}} \in \mathbb{R}^{k^* \times d^*}$. This allows for integrating the information from any combination of modalities, *i.e.*, the modalities in $\mathcal{M}_1, \mathcal{M}_2$ or $\mathcal{M}^*$.

To ensure the output unimodal features $\hat{\mathbf{F}}_m$ are in a common space, we apply a feature alignment loss $L_{\text{align}}$, to all modalities in $\mathcal{M}_1$ and $\mathcal{M}_2$. Specifically, we additionally learn a series of tokens $[\mathbf{u}_1, \cdots, \mathbf{u}_{n_u}], \mathbf{u}_i \in \mathbb{R}^{d^*}$ and encourage the average feature $\bar{\mathbf{f}}_m = \sum_i \hat{\mathbf{f}}_m^i$ of modality $m$ to be close to one learnable token for each sample by minimizing their Euclidean distance. For example, in a classification task, we learn one learnable token per class and encourage the average feature $\hat{\mathbf{f}}_m$ to be close to the corresponding token. For other tasks, we simply encourage the average feature to be close to its nearest learnable token. Thus, for a sample in $\mathcal{X}_{\mathcal{M}_1}$, we formulate our feature alignment loss as:

$$L_{\text{align}} = \sum_{m \in \mathcal{M}_1} ||\bar{\mathbf{f}}_m - \mathbf{u}_{n_m}||_2^2, \tag{1}$$

where $\mathbf{u}_{n_m}$ is the selected learnable token of the current sample for modality $m$. Features from different modalities can be assigned to the same learnable token $\mathbf{u}_t$, when $\mathbf{u}_t$ is the nearest token for both modalities. With the shared learnable tokens, we use the average feature to ensure features occupy a common space while still allowing for modality-specific information.

**Dual Branch Prediction**. From the multimodal features, we could directly learn to make predictions using only groundtruth labels, as in existing multimodal learning methods [28, 9, 32]. However, we find this strategy to be ineffective for modality-incomplete training data, as some modality combinations are less discriminative for the target task. For example, for activity recognition in video, the RGB modality is often more discriminative than audio and optical flow. For the activity *wash carrot*, the sound of water indicates the verb could be *wash* or *rinse*, but the motion of hands cannot distinguish the object, which could be *courgette*, *carrot*, *fruit*, *etc*. In contrast, the RGB modality can make fine-grained distinctions via the appearance. By forcing the model to predict the activity *wash carrot* with audio and optical flow, it may overfit to some unrelated features such as background noise. To mitigate the overfitting of less discriminative modalities, we propose to generate pseudo-supervision which reflects the discriminability of a modality combination and incorporates it alongside the groundtruth labels. To allow separate predictions for the groundtruth and pseudo-supervision, we adopt dual branches with each trained by one type of supervision. We reduce the number of parameters by letting the two branches share the model parameters but have different input features. Specifically, we divide the multimodal feature representation $\hat{\mathbf{F}}$ into two groups, *i.e.*, $\hat{\mathbf{F}}_{\text{gt}} \in \mathbb{R}^{\frac{k^*}{2} \times d^*}$ and $\hat{\mathbf{F}}_{\text{pseudo}} \in \mathbb{R}^{\frac{k^*}{2} \times d^*}$, to form the inputs of the dual branches. Since the two groups of features are trained with different supervision, they learn to capture different semantic information from the input. To reduce the computation cost, we avoid using multiple forward passes and instead send the whole representation $\hat{\mathbf{F}}$ into the transformer layers $\mathcal{H}(\cdot)$ where the attention operations are masked to obtain the prediction of each branch separately. Specifically, for each layer, the output attention becomes:

$$a_{ij} = \frac{I_{ij} \exp\left(\frac{q_i^{\mathsf{T}} k_j}{\sqrt{D}}\right)}{\sum_{\{j', I_{ij'}=1\}} \exp\left(\frac{q_i^{\mathsf{T}} k_{j'}}{\sqrt{D}}\right)}, \tag{2}$$

where $q$ and $k$ indicate the query and key, $i$ and $j$ indicate the $i$th and $j$th tokens in the query and key, and $D$ indicates the dimension of the key. We set $I_{ij}=1$ if $j$ is in the same group as $i$, otherwise $I_{ij}=0$. As a result, we obtain the outputs of the dual branches by a single forward pass.

For the branch trained using groundtruth labels, we simply adopt the standard cross-entropy loss for classification, a triplet loss for retrieval and an L2 loss for regression. For the pseudo-labels used to train the second branch, we average the unimodal predictions across training epochs to get modality-specific pseudo-labels. We obtain the pseudo-labels in such a way as the correctness of predictions of each sample across training epochs can indicate its difficulty [35]. For example, if a sample is often misclassified, it may be too difficult for the available modalities to accurately classify this sample, *i.e.*, there is ambiguous information for recognition. By using average predictions as pseudo-labels to provide a distribution over classes, the model is able to incorporate important distinguishing information while avoiding overfitting as it allows uncertainty between multiple classes. When $\mathcal{M}_1$ or $\mathcal{M}_2$ contain multiple modalities, we select the modality-specific pseudo-label closest to the groundtruth annotation with respect to the cosine similarity. For the pseudo-labels, we adopt the KL divergence loss for classification tasks, while we keep the triplet loss for retrieval and the L2 loss for regression. During testing, we average predictions across branches to obtain the final prediction.

**Overall Training Objective**. For each sample in $\mathcal{X}_{\mathcal{M}_1}$ and $\mathcal{X}_{\mathcal{M}_2}$, our training objective contains three parts: (1) feature alignment loss $L_{\text{align}}$ to ensure that the features of all modalities are projected into the common space with our feature reorganization module; (2) supervised loss $L_{\text{supervised}}$ for the branch using label annotations; (3) pseudo-supervised loss $L_{\text{pseudo}}$ for the other branch using pseudo-labels. The overall training objective is:

$$L = \lambda L_{\text{align}} + L_{\text{supervised}} + \alpha L_{\text{pseudo}}, \tag{3}$$

where $\lambda$ and $\alpha$ are hyperparameters to balance the weights of the loss terms. Since we have multiple training sets for learning, we form batches using randomly selected samples from all training sets.

## 5 Evaluating Unseen Modality Interaction

Since unseen modality interaction is a new problem, we repurpose and reorganize existing multimodal datasets [6, 39, 18] which contain a variety of modalities, as summarized in Table 1.

**Video Classification**. For classification, we use action recognition in EPIC-Kitchens-100 [6] by Damen *et al*. This dataset contains 75,820 first-person video clips captured in 32 different environments. There are three available modalities: RGB, optical flow, and audio. To use this dataset for

| Task | Training Set 1 | Training Set 2 | Validation | Testing |
|---|---|---|---|---|
| Video Classification [6] | RGB | Audio | RGB, Audio | RGB, Audio |
| | RGB | Flow | RGB, Flow | RGB, Flow |
| | Audio | Flow | Audio, Flow | Audio, Flow |
| Robotic State Regression [18] | Image, Depth | Proprioception, Force | All | All |
| Multimedia Retrieval [39] | RGB, Object, Scene, Face | RGB, Audio, OCR, Speech | All | All |

Table 1: **Unseen modality interaction benchmarks** for video classification, robotic state regression and multimedia retrieval with the datasets and modalities per split.

unseen modality interaction, we first divide it into training, validation and testing so that each contains different environments. Training has 62,429 samples, validation 6,750 and testing 6,641. For the data in validation and testing, two modalities are used. The training set is further divided into two equal partitions, each with a different modality and different environments. This results in three settings with different modality combinations: RGB & audio, RGB & optical flow, and audio & optical flow.

**Robot State Regression**. For regression, we use the robotics dataset by Lee *et al*. [18]. The goal is to predict future end-effector positions. The dataset contains 100K states of a real robot collected in manipulation tasks, each with RGB, depth, force, and proprioception modalities. For unseen modality interactions, the training set has 105,554 states, and we divide it into two splits, each containing 52,577 samples. One split has image and depth modalities, while the other contains proprioception and force. Validation and testing contain 20,874 and 17,738 samples and use all four modalities.

**Multimedia Retrieval**. For retrieval, we use the video-text retrieval dataset MSR-VTT [39] by Xu *et al*. This contains 10K YouTube videos with 200K text descriptions. The dataset has seven available modalities [20]: RGB video, audio, object, scene, face, OCR and speech. To evaluate unseen modality interactions, we use the original validation and testing sets with all available modalities but divide the training set into two splits. Since many of the modalities are noisy or missing for some samples, we use the RGB modality in both splits. The first training split contains the 1,702 samples with audio, OCR, speech and RGB. The second split contains the remaining 4,811 video samples with RGB, object, scene and face. The validation and test sets have 127 and 765 video samples respectively.

**Evaluation Criteria**. Following standard practice [6, 10, 20], we report the top-1 accuracy for classification and the mean absolute error (MAE) for regression. For retrieval, we summarize the results with mean rank averaged between video-to-text and text-to-video retrieval in the main paper, we also report full recall, median rank and mean rank metrics in the supplementary.

# 6  Results

**Implementation Details**. For video classification, we use Swin-T [21] pre-trained by Girdhar *et al*. [11] for the RGB unimodal encoder, ResNet [13, 8] for the optical flow encoder, and AST [12] for the audio encoder. For robotic state regression, we use the same feature extractors as in [18], *i.e.*, a series of convolution layers for image and depth and fully connected layers for proprioception and force. For multimedia retrieval, we directly use the unimodal features provided by Liu *et al*. [20].

Both of our feature projection module and transformer layers for prediction consist of six transformer layers [7], each with 8 heads and hidden dimension $d^*$ of 256. The number of learnable tokens for the feature alignment loss is set to $n_u$=3806 on EPIC-Kitchens, and $n_u$=128 on MSR-VTT and the robot dataset. The length of the feature tokens after feature projection is set as $k^*$=512 on EPIC-Kitchens, and $k^*$=16 on MSR-VTT and the robot dataset. We obtain the pseudo-labels by averaging the predictions from the last $e$ epochs of the pretrained unimodal encoders. For video classification, $e$=10. For robot state regression and multimedia retrieval, $e$=20. For the overall training objective (Eq. 3), we set $\alpha$=3000 for video classification, while $\alpha$=1 for regression and retrieval. $\lambda$=0.001 is used for all the three tasks. We use three NVIDIA RTX A6000 GPUs to train our model with a batch size of 96 for video classification, while a single NVIDIA RTX 2080Ti GPU with a batch size of 128 for robotic state regression and multimedia retrieval. Our method is trained with 120 epochs on video classification with a learning rate of $10^{-4}$, reduced to $10^{-5}$ for the last 50 epochs. On robot state regression and multimedia retrieval, we train our method with 50 epochs and a learning rate of $10^{-2}$.

|  | RGB & Audio | RGB & Flow | Audio & Flow | *Mean* |
|---|---|---|---|---|
| **Unimodal** | | | | |
| RGB | 18.2 | 18.2 | - | 18.2 |
| Flow | - | 8.6 | 8.6 | 8.6 |
| Audio | 10.9 | - | 11.8 | 11.4 |
| **Multimodal** | | | | |
| Late fusion | 20.2 | 20.7 | 13.2 | 18.0 |
| Vanilla transformer | 19.3 | 20.1 | 12.2 | 17.2 |
| *This paper:* **Unseen Modality Interaction** | | | | |
| Feature projection | 23.4 | 24.0 | 16.6 | 21.3 |
| + Dual branch prediction | 25.7 | 26.2 | 19.3 | 23.7 |

Table 2: **Ablation of model components** for video classification. We report the top-1 accuracy (%). Note that our proposed feature projection and dual branch prediction are added on top of a vanilla multimodal transformer. Both components contribute to performance improvement for unseen modality combinations and the improvement over a vanilla transformer is considerable.

| $L_{align}$ | $L_{pseudo}$ | RGB & Audio | RGB & Flow | Audio & Flow | *Mean* |
|---|---|---|---|---|---|
| | | 22.1 | 23.2 | 15.1 | 20.2 |
| ✓ | | 23.5 | 24.2 | 16.3 | 21.4 |
| ✓ | ✓ | 25.7 | 26.2 | 19.3 | 23.7 |

Table 3: **Ablation of training objective** using video classification. We report the top-1 accuracy (%). Both loss terms improve the generalization to unseen modality interactions.

**Ablation of Model Components**. We ablate the effect of our two proposed model components, *i.e.*, feature projection and dual branch prediction, on our unseen modality interaction for video classification. We compare with: (i) *Unimodal encoders* which use the unimodal features of a single modality with a linear layer for prediction, (ii) *Late fusion* which uses the average of the predictions from the unimodal encoders, (iii) *Vanilla transformer* which uses the multimodal fusion from [9]. To use this vanilla transformer with modality-incomplete training data, we use a series of per-modality learnable tokens in place of the missing modality. The results are shown in Table 2. Note that for each column, models only see unimodal data in training and we consider the unseen combination of both modalities at inference. Combining unimodal features by either late fusion or a vanilla transformer is beneficial for all modality combinations as different modalities provide complementary information. However, using the vanilla transformer is worse than simply averaging the predictions for our task of unseen modality interactions, since the vanilla transformer cannot learn the cross-modal correspondences among different modalities without modality-complete data. By projecting the features from different modalities into a common space before applying the vanilla transformer, our model obtains a 4.1% average improvement over the vanilla transformer. This demonstrates that combining modality-specific features in the common space improves the generalizability to unseen modality combinations. Adopting dual branch prediction in the transformer delivers a further 2.4% improvement on average. This highlights the importance of using pseudo-supervision to avoid overfitting to modalities. Overall, our model achieves a top-1 accuracy of 23.7% on average with modality-incomplete training data. When considering an oracle with all training data for all modalities available, the accuracy would be 26.1%. Thus, our model closes the gap with the oracle considerably.

**Ablation of Training Objective**. Our training objective in Eq. 3 consists of three terms. We ablate the effectiveness of adding the alignment loss $L_{align}$ and the pseudo-supervised loss $L_{pseudo}$ to the task-specific loss. We use our full model with both the feature projection and the dual branch prediction. When not using the pseudo-supervised loss, we use the same supervised loss for both branches. Table 3 shows the results. The +1.2% improvement when adding our feature alignment loss indicates that this loss facilitates the projection of different modalities into a common space so that the model can better generalize towards unseen modality combinations. Our pseudo-supervised loss further improves the accuracy by +2.4% as it eliminates the overfitting to groundtruth labels, which can be harmful when a particular modality combination cannot give a reliable prediction.

**Ablation of Number of Parameters**. Since our feature projection module introduces additional parameters over a vanilla multimodal transformer [9], we further demonstrate that the benefit of our model comes from our proposals, not the additional parameters used. Specifically, we reduce the

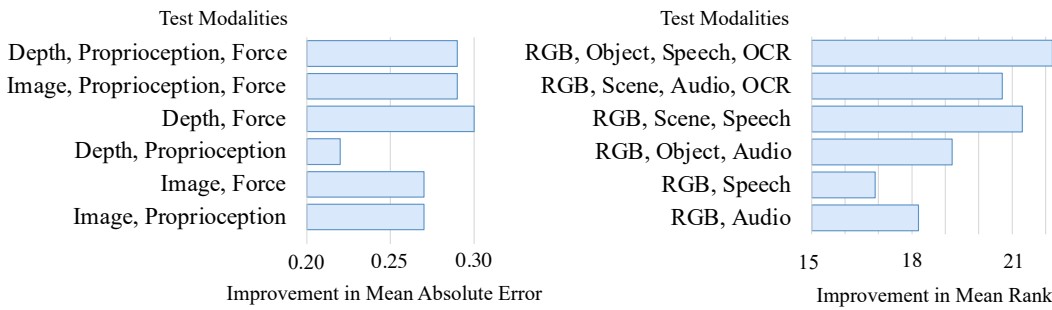

(a) Robotic State Regression                    (b) Multimedia Retrieval

Figure 3: **Benefit for Modality-Incomplete Testing** with robot state regression and multimedia retrieval. The modality combinations used in training are the same as Table 1. We show the improvement over a vanilla multimodal transformer. Our model improves the robustness for all unseen modality combinations.

number of parameters from 14.4M to 7.2M, which is equal to the number used in a vanilla multimodal transformer. With an equal number of parameters, our model achieves 21.4% on EPIC-Kitchens, still outperforming the vanilla counterpart by 4.2%. Thus, the improvement of our model is due to the effective method for learning unseen modality interactions.

**Alternative Feature Projection**. Previous works [24, 27] also consider projecting features of different modalities into a common space. However, they only project the feature vector for the final prediction, instead of the multidimensional features, which contain much richer information. To further demonstrate the effectiveness of our feature projection, we replace it with an average pool on the multidimensional features, followed by a linear layer as in previous methods. Then, we obtain 19.0% on EPIC-Kitchens with RGB, optical flow, and audio compared to 23.7% by our method. Thus, our feature projection module is more effective as it keeps rich information for accurate prediction.

**Beyond Two Unimodal Subsets**. Up to this point, we have learned our model using two training sets with distinct modalities. Here we confirm our model extends beyond this to more distinct training and modality sets. We test this with video classification, where we partition the training set into three equal splits, with RGB, optical flow and audio respectively. We keep the testing and validation sets unchanged but use all three modalities. We obtain 23.8% top-1 accuracy, surpassing the late fusion with 18.4% and the vanilla transformer with 17.5%. Thus, our model can solve the general unseen modality interaction task where there are more than two training splits with distinct modalities.

**Benefit for Modality-Incomplete Testing**. While our model learns from modality-incomplete data, our experiments up to this point use modality-complete data in testing. Here we investigate whether our model is also beneficial to modality-incomplete data at inference. We test this for robot state regression and multimedia retrieval as these have many modalities. Figure 3 shows the improvement of our approach over a vanilla transformer for six different modality combinations for each task. Our model can handle any combination of input modalities at inference and improves over the vanilla transformer for all unseen combinations. We also note that our model is most effective when combining a larger number of modalities as this provides more diverse information.

**Benefit for Noisy Modalities**. We further demonstrate the benefit of our model by testing it on samples with noisy modalities for unseen modality interactions with video classification. To this end, we apply Gaussian noise on one of the modalities for all samples across the test set. While late fusion delivers 11.2% accuracy, the vanilla multimodal transformer shows a worse accuracy of 10.1%, since the noise makes cross-modal correspondences harder to find. In contrast, our model is more robust to noise and obtains 18.0%. Our model's robustness comes from the feature projection module which selects the informative features during the projection and can thus reduce the noise.

**Comparative Results**. As there are no prior methods for unseen modality interaction, we compare our approach with recent multimodal learning methods which assume all data to be modality-complete: Gabeur *et al*. [9], Nagrani *et al*. [28], Wang *et al*. [36], as well as methods which are robust to some modality-incomplete data: Recasens *et al*. [29] and Shvetsova *et al*. [32]. We use publicly available implementations, except for the work by Recasens *et al*. [29], which we re-implement

| | Video Classification | Robot State Regression | Multimedia Retrieval |
|---|---|---|---|
| | Top-1 (%) ↑ | MAE ↓ | MnR ↓ |
| Late fusion[‡] | 18.1 | 1.29 | 72.3 |
| **Modality Complete** | | | |
| Gabeur *et al.* | 17.3 | 1.37 | 88.8 |
| Nagrani *et al.* | 17.5 | 1.35 | 86.2 |
| Wang *et al.* | 17.3 | 1.37 | 87.8 |
| **Modality Incomplete** | | | |
| Shvetsova *et al.* | 18.0 | 1.25 | 72.3 |
| Recasens *et al.* [†] | 18.5 | 1.22 | 72.2 |
| **Unseen Modality Interaction** | | | |
| *This paper* | **23.7** | **1.07** | **66.2** |

[‡] Averaging the final predictions from unimodal encoders.
[†] Based on our re-implementation.

Table 4: **Comparison with multimodal learning methods** for video classification, robot state regression and multimedia retrieval tasks. While previous multimodal learning methods need modality-complete data to learn the cross-modal correspondences, our method gives more effective cross-modal fusion for unseen modality combinations.

ourselves. For all models, we use the same unimodal feature extractors as ours. To adapt modality-complete methods [9, 28, 36] to the unseen modality interaction, we learn a series of learnable feature tokens for each modality and use these learnable tokens in place of the missing modality. Table 4 shows the results for classification, regression, and retrieval tasks. We find that modality-complete methods perform worse than late fusion as they cannot learn the cross-modal correspondences without modality-complete data, therefore cannot effectively make predictions for unseen modality combinations during inference. While the modality-incomplete approaches are robust to some missing modality data, they are not for unseen modality interactions where there is no modality-complete data in training. Thus, we outperform these approaches by effectively accumulating the information from any modality combinations. Overall, existing approaches are not robust to unseen modality combinations whereas our model obtains better generalization ability from modality-incomplete training data. It does so for many different modalities including RGB, audio, depth, and force, and a diverse set of tasks including video classification, robot state regression, and multimedia retrieval.

## 7 Discussion

**Conclusion**. We pose the problem of unseen modality interaction, where the modality combinations that appear during inference have not been seen during training. To facilitate further analysis and progress, we repurpose and reorganize three existing multimodal datasets to evaluate our approach on a wide variety of modalities, domains, and tasks. Experiments show that we can effectively make predictions for unseen modality interactions by accumulating different modalities projected in a common space and incorporating pseudo-supervision indicating the reliability of the multimodal representation. Our approach is suitable for classification, regression, and retrieval, can handle a wide variety of modality combinations, and is more robust to noisy modalities.

**Limitations**. Most of our experiments use equally sized training subsets. The effect of an imbalance in the number of samples or number of modalities in different training subsets would be interesting to examine. While effective, our method comes with a longer training time. We also explore only fully supervised learning. Future work should tackle the extension to a self-supervised learning framework.

**Broader impact**. Learning from data with missing modalities is important for machine learning, as collecting sufficient training data with all modalities of interest can be difficult. A multimodal model that can make predictions with unseen modality combinations can increase robustness when applied to real-world applications. Learning unseen modality interaction is orthogonal to other machine learning challenges like continual learning, federated learning, as well as domain adaptation and generalization, whose combination opens up new lines of research.

**Acknowledgement.** This work is financially supported by the Inception Institute of Artificial Intelligence, the University of Amsterdam and the allowance Top consortia for Knowledge and Innovation (TKIs) from the Netherlands Ministry of Economic Affairs and Climate Policy.

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
