| Method | Text $\implies$ Video | | | | | Video $\implies$ Text | | | | |
|---|---|---|---|---|---|---|---|---|---|---|
| | R@1 ↑ | R@5 ↑ | R@10 ↑ | MdR ↓ | MnR ↓ | R@1 ↑ | R@5 ↑ | R@10 ↑ | MdR ↓ | MnR ↓ |
| Late fusion | 5.2 | 16.9 | 27.7 | 32.0 | 72.1 | 5.4 | 17.6 | 26.5 | 31.0 | 72.4 |
| **Modality Complete** | | | | | | | | | | |
| Gabeur *et al.* | 3.5 | 15.8 | 26.1 | 35.0 | 76.5 | 2.5 | 11.2 | 18.6 | 47.0 | 101.1 |
| Nagrani *et al.* | 3.8 | 16.1 | 25.6 | 35.0 | 75.2 | 2.8 | 12.1 | 18.6 | 46.0 | 97.2 |
| Wang *et al.* | 3.5 | 16.1 | 25.9 | 35.0 | 76.6 | 3.1 | 11.6 | 19.0 | 46.0 | 98.9 |
| **Modality Incomplete** | | | | | | | | | | |
| Shvetsova *et al.* | 5.1 | 16.9 | 27.5 | 33.0 | 72.5 | 5.6 | 17.2 | 25.9 | 31.0 | 72.0 |
| Recasens *et al.* [†] | 5.0 | 17.1 | 28.2 | 33.0 | 73.2 | 5.2 | 17.4 | 26.5 | 30.0 | 71.2 |
| **Unseen Modality Interaction** | | | | | | | | | | |
| *This paper* | 6.0 | 19.2 | 30.5 | 28.0 | 66.4 | 6.3 | 18.9 | 29.0 | 27.0 | 65.9 |

[†] We re-implement the method ourselves.

Table 5: **Comparison with multimodal learning methods** for the retrieval task using all metrics for MSR-VTT. While multimodal learning methods need modality-complete data to learn the cross-modal correspondences, our method gives more effective cross-modal fusion for unseen modality combinations.

## A  Comparative Results for Retrieval

In Table 4 in the main paper, we summarized the multimedia retrieval results with the Mean Rank (MnR) averaged between video-to-text and text-to-video. Here, we provide the full comparison with all the metrics for both text-to-video retrieval and video-to-text retrieval in Table 5. As recent multimodal learning methods need modality-complete data for training, our model outperforms these approaches on all metrics by effectively accumulating the information from any modality combinations.

## B  Hyperparameter Analysis

Here, we study the effect of hyperparameters used in our model, including the length of feature tokens $k^*$ after feature projection, the number of learnable tokens $n_u$ for feature alignment loss, $\alpha$ and $\lambda$ to balance the loss terms.

**Number of Learnable Tokens $n_u$ for the Feature Alignment Loss**. When learning our feature projection module, we ensure features of different modalities are projected into a common space by applying a feature alignment loss with $n_u$ learnable tokens. While $n_u$ can be simply equal to the number of classes for classification task, it serves as a hyperparameter for the regression and retrieval tasks. We ablate the effect of $n_u$ in Table 6a using the retrieval task. With fewer learnable tokens, the projected features become less discriminative since many of them with different semantic meanings need to match the same learable token. Increasing $n_u$ to 128 is effective after which the model is relatively robust to the choice of this hyperparameter.

**Length of Feature Tokens $k^*$**. Our feature projection module projects modality-specific features into a common space with $k^*$ feature tokens for each modality. We investigate the impact of the size of $k^*$ using video classification in Table 6b. With a smaller $k^*$, the module cannot reserve all the discriminative features for multimodal recognition. However, with a larger $k^*$, the module tends to also reserve the unimportant features, which result in overfitting. Thus, our model achieves good performance when $k^*$ is in the range of 256 and 1024.

**Token Partition in Dual Branch Prediction.**  In the main paper, we divide the tokens in half for the dual branches. Here, we test the model's performance with different partition strategies on EPIC-Kitchens and report the results in Table 6c. Dividing the tokens into half delivers the best performance. While the pseudo-labels can refine the overconfident predictions trained by groundtruth labels, they may also be noisy for some difficult samples while the groundtruth labels provide the correct supervision. Thus, making the pseudo supervision and the groundtruth supervision equally important is beneficial.

| $n_u$ | MnR $\downarrow$ |
|---|---|
| 32 | 73.2 |
| 64 | 69.3 |
| 128 | 66.2 |
| 256 | 66.9 |
| 512 | 68.4 |

(a) Effect of $n_u$

| $k^*$ | Top-1 (%) $\uparrow$ |
|---|---|
| 128 | 20.9 |
| 256 | 22.3 |
| 512 | 23.7 |
| 1024 | 22.7 |
| 2048 | 21.2 |

(b) Effect of $k^*$

| Ratio of $L_{\text{pseudo}}$ to $L_{\text{supervised}}$ | Top-1 (%) $\uparrow$ |
|---|---|
| 30:70 | 22.9 |
| 50:50 | 23.7 |
| 70:30 | 22.5 |

(c) Token Partition for Dual Branch Prediction

| $\alpha$ | Top-1 (%) $\uparrow$ |
|---|---|
| 0 | 21.4 |
| 2000 | 21.9 |
| 2500 | 22.8 |
| 3000 | 23.7 |
| 3500 | 22.5 |
| 4000 | 21.0 |

(d) Effect of $\alpha$

| $\lambda \times 10^{-3}$ | Top-1 (%) $\uparrow$ |
|---|---|
| 0.0 | 21.7 |
| 0.1 | 22.2 |
| 0.5 | 22.7 |
| 1.0 | 23.7 |
| 1.5 | 22.9 |
| 2.0 | 22.0 |

(e) Effect of $\lambda$

Table 6: **Effecf of Hyperparameters.** Note that (a) uses the multimedia retrieval task, while others use the video classification task. (a) Increasing $n_u$ to 128 is effective, then performance plateaus. (b) Projecting modality-specific features into more tokens results in overfitting, while projecting into less tokens leads to underfitting. With $k^*$=512, we obtain the best trade-off. (c) While the pseudo-labels can either be noisy or refine the overconfident predictions, dividing the tokens into half delivers the best trade-off. (d) For $\alpha$, any value in the range of 2500 and 3500 results in a good trade-off between underfitting to the pseudo-labels and overfitting. (e) $\lambda$=$10^{-3}$ delivers the best trade-off between feature alignment and target task prediction. Default settings are shaded in  gray .

**Effect of $\alpha$ and $\lambda$ to Balance the Loss Terms**. In Eq. 3, we set $\alpha$=3000 and $\lambda = 0.001$ for video classification to balance the losses among pseudo supervision, feature alignment and groundtruth supervision. Here, we ablate their effects in Table 6d and Table 6e using the video classification task. For $\alpha$, any value in the range of 2500 and 3500 results in a good trade-off between underfitting to the pseudo-labels and overfitting, and the performance improvement over the counterpart without $L_{\text{pseudo}}$ (*i.e.*, $\alpha = 0$) is considerable. For $\lambda$, with a lower value, the features from different modalities cannot be aligned effectively, while a larger $\lambda$ makes the model focus less on learning the target prediction task. Thus, $\lambda$=$10^{-3}$ delivers the best trade-off.

## C  Ablation

**Feature distance with the feature alignment loss $L_{\text{align}}$.** The proposed feature alignment loss in Eq. 1 aims to facilitate the projection of features from distinct modalities into a shared space. We further verify this claim by computing the feature distance between modalities on EPIC-Kitchens with the variants of our model used in Table 2 of the main paper. Specifically, we first obtain the average feature before the multimodal transformer per class for each modality and then compute the average Euclidean distance between modalities across classes. For RGB & Audio, after adding the feature projection to the vanilla transformer, the average Euclidean distance reduces from 84.1 to 75.4. For RGB & Flow, the distance reduces from 81.3 to 72.5 and for Audio & Flow, it drops from 83.9 to 76.2. Thus, our feature alignment loss does encourage features of different modalities to be projected into a common feature space.

**Robustness to noise.** In the main paper, we show that when applying Gaussian noise $\mathcal{N}(0, 1)$ on one modality for all test samples in video classification, our method is more robust to the noise than late fusion or a vanilla multimodal transformer. In Table 7, we further impose different amounts of noise and compare our approach with recent multimodal learning methods which assume all data to be modality-complete: Gabeur *et al.* [9], Nagrani *et al.* [28], Wang *et al.* [36], as well as methods which are robust to some modality-incomplete data: Recasens *et al.* [29] and Shvetsova *et al.* [32].

| Model | $\mathcal{N}(0, 0.5)$ | $\mathcal{N}(0, 1)$ | $\mathcal{N}(0, 2)$ |
|---|---|---|---|
| Late fusion | 14.1 | 11.2 | 10.0 |
| Gabeur *et al.* | 13.2 | 10.1 | 9.3 |
| Nagrani *et al.* | 16.4 | 15.5 | 14.4 |
| Wang *et al.* | 14.9 | 12.3 | 10.8 |
| Shvetsova *et al.* | 15.5 | 13.4 | 11.3 |
| Recasens *et al.* | 15.3 | 13.1 | 12.0 |
| *This paper* | **20.5** | **18.0** | **16.2** |

Table 7: **Robustness to Noise**. Our model achieves better results when modalities are corrupted by severe noise than these prior works.

| Model | Gabeur *et al.* | Nagrani *et al.* | Wang *et al.* | Shvetsova *et al.* | Recasens *et al.* | *This paper* |
|---|---|---|---|---|---|---|
| RGB, Audio, OCR, Speech | 90.6 | 89.7 | 90.0 | 90.6 | 90.5 | **79.4** |
| RGB, Object, Scene, Face | 89.1 | 88.8 | 88.9 | 89.3 | 89.4 | **80.3** |
| RGB, Object, Speech, OCR | 92.4 | 90.2 | 91.5 | 78.1 | 77.4 | **70.2** |
| RGB, Scene, Audio, OCR | 91.0 | 89.9 | 90.7 | 75.3 | 74.2 | **70.3** |
| RGB, Scene, Speech | 95.6 | 94.5 | 95.1 | 80.2 | 79.3 | **74.3** |
| RGB, Object, Audio | 96.1 | 96.0 | 96.3 | 82.1 | 80.3 | **76.9** |
| RGB, Speech | 98.3 | 98.0 | 98.8 | 84.6 | 83.0 | **81.4** |
| RGB, Audio | 98.0 | 97.3 | 98.4 | 85.3 | 84.5 | **79.8** |

Table 8: **Benefit for Modality-Incomplete Testing** with multimedia retrieval. Performance is reported in the mean rank metric, for which the lower the better. While works assuming modality-complete data overfit to seen combinations (top two rows), those dealing with some modality-incomplete training data are not generalizable to all combinations. In contrast, our method is the most effective on all seen and unseen modality combinations.

| Model | Gabeur *et al.* | Nagrani *et al.* | Wang *et al.* | Shvetsova *et al.* | Recasens *et al.* | *This paper* |
|---|---|---|---|---|---|---|
| Image, Depth | 1.40 | 1.39 | 1.41 | 1.38 | 1.41 | **1.29** |
| Force, Proprioception | 1.39 | 1.37 | 1.39 | 1.37 | 1.39 | **1.27** |
| Depth, Proprioception, Force | 1.48 | 1.45 | 1.44 | 1.37 | 1.39 | **1.27** |
| Image, Proprioception, Force | 1.47 | 1.43 | 1.45 | 1.32 | 1.30 | **1.18** |
| Depth, Force | 1.77 | 1.72 | 1.79 | 1.62 | 1.58 | **1.47** |
| Depth, Proprioception | 1.60 | 1.52 | 1.55 | 1.49 | 1.43 | **1.38** |
| Image, Force | 1.71 | 1.64 | 1.70 | 1.58 | 1.52 | **1.44** |
| Image, Proprioception | 1.54 | 1.50 | 1.52 | 1.38 | 1.35 | **1.27** |

Table 9: **Benefit for Modality-Incomplete Testing** with robot state regression. The performance is reported in the mean absolute error (MAE), for which the lower the better. While works assuming modality-complete data overfit to seen combinations (top two rows), those dealing with some modality-incomplete training data are not generalizable to all combinations. In contrast, our method is the most effective on all seen and unseen modality combinations.

We conclude that our model achieves better results when modalities are corrupted by severe noise than these prior works.

**Reducing overfitting to specific modality combinations**. In Figure 3 in the main paper, we show our model improves robustness to various unseen modality combinations over a vanilla multimodal transformer. To further demonstrate our generalizability to different modality combinations and demonstrate that prior works indeed overfit to the combinations in training, we expand this experiment. Specifically, we compare our method on both seen and unseen modality combinations with recent multimodal learning methods which assume all data to be modality-complete: Gabeur *et al.* [9], Nagrani *et al.* [28], Wang *et al.* [36], as well as methods which are robust to some modality-incomplete data: Recasens *et al.* [29] and Shvetsova *et al.* [32]. The results are reported in Table 8 and Table 9. Since Gabeur *et al.* [9], Nagrani *et al.* [28] and Wang *et al.* [36] assume modality-complete data, they obtain worse performance with unseen combinations than with seen combinations (top two rows), even with additional modalities. Thus we conclude these methods overfit to seen combinations. Since Shvetsova *et al.* [32] and Recasens *et al.* [29] aim to be robust to some modality-incomplete training data, they benefit from some modality combinations. However, these methods are not generalizable to all combinations. In contrast, our method is the most effective on all seen and unseen modality combinations.

# D Unseen Modality Interaction Benchmarks

In the main paper, we present the modalities available in each split for the three unseen modality interaction benchmarks 1. Here, we list the number of samples in each split in Table 10.

| Task | Training Split 1 | | Training Split 2 | | Val | Testing |
|---|---|---|---|---|---|---|
| | Modalities | #Samples | Modalities | #Samples | | |
| Video Classification [6] | RGB | 31,213 | Audio | 31,216 | 6,750 | 6,641 |
| | RGB | 31,213 | Flow | 31,216 | | |
| | Audio | 31,213 | Flow | 31,216 | | |
| Robotic State Regression [18] | Image, Depth | 52,577 | Proprioception, Force | 52,577 | 20,874 | 17,738 |
| Multimedia Retrieval [39] | RGB, Object, Scene, Face | 4,811 | RGB, Audio, OCR, Speech | 1,702 | 127 | 765 |

Table 10: **Unseen modality interaction benchmarks** for video classification, robotic state regression and multimedia retrieval with the datasets, modalities and number of samples per split.