# OpenReview forum: "Learning Unseen Modality Interaction"
_NeurIPS.cc/2023/Conference — NeurIPS 2023 poster_

### Official Review · Reviewer_d7nB · 2023-06-28

**Soundness:** 4 excellent
**Presentation:** 2 fair
**Contribution:** 3 good
**Rating:** 6
**Confidence:** 3

**Summary:**

This work studies the problem of learning interactions of unseen modality combinations. Specifically, all training data is modality-incomplete, and the model must learn to perform inference on modality-complete data. The paper claims to be the first to study inference under such settings, and proposed two novel improvements to tackle this challenge: (1) feature projection layer that projects all encoded modalities into the same dimensionality and constrained by an alignment loss, and (2) a dual-branch prediction layer that predicts a pseudo-label in addition to the real label. The paper performed experiments on 3 datasets that contains a diverse set of modalities, domains and tasks (including classification, retrieval and regression). The paper included thorough ablation studies on their methods to justify all of their design choices, and they showed that under the their new setting, their method significantly outperforms previous modality-complete and modality-incomplete approaches.

**Strengths:**

1. This paper proposed a new setting: learning to infer from unseen combinations of modalities, and proposed a new method to tackle this challenge.

2. The paper's experiments and evaluations are very comprehensive. The experiments involved 3 datasets that contains a very diverse set of different domains (kitchen videos, robotics, Youtube videos), modalities (14 total modalities), and task (classification, regression, retrieval), and the diversity shows that their method generalizes well. There is a comprehensive ablation study that justifies the design choice of each model component as well as loss function. Since this is a new setting, the authors re-implemented or re-ran several existing approaches on the new setting and showed that the new method outperforms all of them.

**Weaknesses:**

The presentation/clarity of the paper needs improvement, especially in some parts of the methods section. For example, the use of different variable names in the feature projection are inconsistent (e.g. F'm  in line 96 and line 102 have different dimensions); the notation of the alignment process is confusing, and the intuition behind the whole alignment process is unclear; and the description of how exactly the psudo-labels are obtained is very vague and unclear. See Questions section for more details.

**Questions:**

(1) I think figure 1 can be a bit misleading, since in this paper, we focused on the setting where the test set is always the union of all modalities.

(2) The dimension of F'm on line 96 and 102 are inconsistent.

(3) For feature projection, is it correct that we simply do a linear combination to change the sequence dimension? When we change dm to d*, do we have a fully connected layer that maps each dm-dimensional vector to a d* dimensional vector individually, or does it map the entire k*x dm matrix to a k* x d* matrix?

(4) On line 111, why do we average the features within each modality? It seems to me that the alignment loss is sort of like a vector-qualitization process, and it is difficult for me to understand why quantizing each modality separately (as opposed to, for example, quantizing each d-dimensional vector in the sequence, or the average of the vectors in the same corresponding locations across different modalities) helps features from different modalities occupy a common space. Wouldn't the current approach prevent the projected features from occupying a common space? For example, we could have modality 1 always close to the first few u in the dictionary, and modality 2 always close to the next few u, etc.

(5) In equation 1, the notation of u_m is a bit confusing. I think it is supposed to mean the closest vector in [u1, u2,... ] to fm, but it could be confused with the mth vector in [u1,u2,...]

(6) What exactly does "average across training epochs" mean? Since we need the pseudo-labels during training, do we just average the first-branch prediction on this data point from each previous epoch? How do we obtain this for the first epoch? Also, is it correct that the pseudo-labels are probability-distributions across all labels in the classification case? I also have a hard time understanding the intuition behind dual-branch prediction helping with overfitting problem. Perhaps a more clear and detailed description on how the pseudo-labels are obtained could make it more clear.

(7) In line 167, since there are 3 modalities in this dataset, how exactly are they partitioned into the two training set partitions?

**Limitations:**

Limitation discussion is adequate.

---

> ### Author Rebuttal · Authors · 2023-08-08
>
> ***We thank the reviewer for their time and effort. We are glad that the review appreciated the new setting of unseen modality interaction, the new method proposed for this and found  the experiments and evaluations comprehensive***.
>
> &nbsp;
>
> **Figure 1.** We will make Figure 1 clearer by presenting two scenarios where all the modalities are available and where subsets of the modalities are available during inference. While our experiments in Tables 1-3 do focus on the test set having the union of all modalities, in Figure 3 we show that our method is beneficial to subsets of  modalities. We will also expand the analysis in Figure 3 in the final version with the comparison to previous multimodal learning methods provided in the response to reviewer 2qSh.
>
> **Dimension of $F’_m$.** The reviewer is correct. We made a mistake here and F’m on line 96 should be modified to $\hat{F}_m$.
>
> **Feature projection.** We have a fully connected layer that maps each dm-dimensional vector to a d dimensional vector individually.
>
> **Averaging the features within each modality.** We average the modality specific features for the alignment loss instead of aligning each individual feature vector as many feature vectors may be uninformative to the target problem. We demonstrate that averaging is better than aligning individual features with an experiment on EPIC-Kitchens. When encouraging each d-dimensional vector to be close to one of the learnable tokens, we obtain 20.7%. When using the individual features averaged across modalities, we get 19.8%. Both are worse than our 23.7%. Thus, our alignment strategy is more effective.
>
> The reviewer is correct that it could be possible to have one modality always close to the first few u in the dictionary and another modality always close to other u in the dictionary. However, we observed that this didn’t happen and therefore didn’t find it necessary to discourage such cases in the alignment loss. Instead, we observe that each modality covers the majority of the tokens across the training samples, allowing the predictions to be diverse and satisfy the groundtruth supervision.
>
> **Notation $u_m$.** Yes, we mean the closest vector in [u1, u2, …] to $\bar{f_m}$. We agree that the notation of $u_m$ is confusing and modify it as:
> $$L_{align} = \sum_{m \in M_1} ||\bar{f_m} - u_{n_m}||^2_2,$$
> where $u_{n_m}$ is the learnable token from $[u_1, …, u_{n_u}]$ selected for feature $\bar{f}_m$.
>
> **Average across training epochs.** We obtain the pseudo-labels by averaging the predictions from the last e epochs of the pretrained unimodal encoders. For video classification e=10, for robot state regression e=20 and for multimedia retrieval e=20. We will add these details to the paper. Yes, the pseudo-labels are probability distributions across all labels in the classification case.
>
> **Intuition behind dual-branch prediction.** Our intuition for the pseudo-labeling strategy is inspired by the observation that a single modality alone often cannot provide enough information for accurate prediction. Take the example of conducting activity recognition with audio and video modalities, the audio is often less discriminative than video. For instance, the audio modality can be crucial in distinguishing that the activity is one of *swimming*, *surfing* or *water skiing*, but cannot make fine-grained distinctions. By forcing the model to predict the ground-truth activity *swimming*, it may overfit to some unrelated features such as background noise. By using average predictions as pseudo-labels to provide a distribution over classes, the model is able to incorporate the important distinguishing information while avoiding such overfitting as it allows uncertainty between multiple classes. We will make this clearer in the final version.
>
> **Training set partitions.** We have three settings on video classification with the EPIC-Kitchens dataset each using two of the three modalities. Therefore, we divide the training set into two splits with each containing only one modality.

---

> > ### Comment · Reviewer_d7nB · 2023-08-15
> >
> > Thank you for your response! My review remains positive.

---

### Official Review · Reviewer_2qSh · 2023-07-04

**Soundness:** 2 fair
**Presentation:** 3 good
**Contribution:** 2 fair
**Rating:** 6
**Confidence:** 2

**Summary:**

The paper is about multimodal learning and deals in particular with the mismatch between modality combinations at training and inference time.
The authors propose to project the multimodal features in a shared space, apply an alignment, and enforce the discriminative ability of the method with a dual branch prediction with full and pseudo supervision.
The method is evaluated on three scenarios, ablation studies and comparative analysis are reported.

**Strengths:**

Overall, the paper is well-written, with some originality with respect to previous approaches. What is reported in the document is clear, figures are appropriate and descriptive of the concepts.

**Weaknesses:**

My main concerns on the paper are the following.
On the main goal: while I understand the benefit of learning from multiple modalities while being able to use a single modality at inference time,  I am not sure I fully understand the benefit of learning according to the set defined in Sect. 2, where at training time the method learns from a group of modalities which is a subset of what is seen at inference time.
I expected to see in the empirical evaluation a quantitative justification that using at inference time a superset of the training modalities helps to improve the results. For instance, what if at inference time I use the combination used at training time, with no additional modalities? Is it really helpful to have these extra modalities at inference time? This is not clear to me.
On the objectives: in my opinion, some somehow strong statements of the authors on the abilities of their method are not appropriately justified with an empirical evaluation. For instance, the authors state that their method is more robust to scenarios where some modalities are corrupted by severe noise. More robust with respect to who and what? There is only one experiment tackling this challenge where only the proposed method is used. Moreover, the authors state that one of their challenges "...is reducing the overfitting to the specific modality combinations from the modality-incomplete training data." Again, I think this would deserve more attention on the comments to the experimental analysis
I am not sure I fully understood how the architecture I structured: I assume the architecture is accommodating a maximum number of input modalities while disregarding (or having learnable tokens for) the ones that are not present. From what I understand this requires the input modalities to be provided always in the same order. Is this interpretation correct?

**Questions:**

- I find it confusing having a separation of the considerations on the existing literature at the beginning and at the end of the manuscript, I would group them
- On the same topic, the authors mention existing approaches and how their method is different. However (at least in the first part) only for some of them ([25, 30]) they explicitly mention limitations, what about the others?
- As reported above: the statement "Our approach enables effective unseen modality interaction and is more robust to scenarios where some modalities are corrupted by severe noise." would require a deeper investigation I think
- What about the ability to generalize to new tasks/settings, different from the ones of the training set? What I mean is: if you have in training data collected with a certain combination of sensors from a certain platform to solve an action classification problem for instance, could the method be able to generalize to test data for the very same problem and possibly different sensors combinations acquired with a different platform?
- The section on method lacks details in the description that are probably then reported at the beginning of the section on experimental analysis. For instance, it is said in Section 3 that the attention matrix Om is obtained through several transformer layers. Be sure all the details are properly reported. This is very important for the reproducibility of the results
- I don’t understand when the alignment is applied, before or after the integration via the sum.
- Eq. (2) appears without the appropriate context, I find
- “… we propose to generate pseudo-supervision which reflects the discriminability a modality combination” Sentence to be rephased
- Are the splittings in the training-validation-test used for the experiments provided with the datasets? If yes it should be stated, if not it should be justified
- How many epochs for the training stage? Learning rate? Again, this is important to share as many implementation details as possible to favour the reproducibility
- References to tables 7 and 8 should be 2 and 3 instead
- “While our model learns from modality-incomplete data, previous experiments use modality-complete data in testing” It would be better to cite the methods
- In the section “Benefit for Modality-Incomplete Testing” it is not quite clear how the experiment has been designed. Are the combinations in Figure 3 used at inference time? What was the training? I think some details are missing. The same considerations can be done for other sections, for instance “Benefit for Noisy Modalities“
- In the experiments: “We use publicly available implementations where available, otherwise re-implementing ourselves.” Please be more specific: for which ones you used public implementations?

**Limitations:**

The authors explicitly mention some limitations of their method in the last section.

---

> ### Author Rebuttal · Authors · 2023-08-08
>
> ***We thank the reviewer for their time and effort and are glad the reviewer found our paper well-written***.
>
> &nbsp;
>
>
> **Benefit of supersetting training modalities.** We show the benefit in Table 2 where we compare our model and multimodal baselines, which use all available modalities, to unimodal results which use the single modalities seen in training. For example, our method achieves 25.7% accuracy with RGB & audio as opposed to 18.2% using only RGB and 10.9% using only audio. We will highlight this. Furthermore, when using only RGB or audio, our method gives 19.6% and 12.3%, worse than using both modalities (25.7%). Thus, using a superset at inference is beneficial.
>
> **Benefit for Modality-Incomplete Testing.** We use the same model as in Table 4: trained on the modality-incomplete data splits as described in Section 4. In Fig 3 we test the model on different modality combinations at inference. For “Benefit for Noisy Modalities”, we also use the same trained model but add noise to some modalities during inference. We will add these clarifications.
>
> **Robustness to noise.**. In our paper, we show that when applying Gaussian noise $N(0,1)$ on one modality for all test samples in video classification, our method is more robust to the noise than late fusion or a vanilla multimodal transformer. We further compare our approach with more methods under different amounts of noise:
> |Model|N(0,0.5)|N(0,1)|N(0,2)|
> |-|-|-|-|
> |Late fusion|14.1|11.2 |10.0|
> |Gabeur et al.|13.2|10.1|9.3|
> |Nagrani et al.|16.4|15.5|14.4|
> |Wang et al.|14.9|12.3|10.8|
> |Shvetsova et al.|15.5|13.4|11.3|
> |Recasens et al.|15.3|13.1|12.0|
> |Ours|**20.5**|**18.0**|**16.2**|.
> We conclude that our model achieves better results when modalities are corrupted by severe noise than these prior works and will add this table.
>
> **Reducing overfitting to specific modality combinations.** In Fig 3, we show our model improves robustness to various unseen modality combinations over a vanilla multimodal transformer. To further demonstrate our generalizability to different modality combinations and demonstrate that prior works indeed overfit to the combinations in training, we expand this experiment. Specifically, we compare our method on both seen and unseen modality combinations with previous fusion works and report the results in the tables below. Note for both mean rank (multimedia retrieval) and MAE (robot state regression), lower is better.
> |Model|Gabeur et al.|Nagrani et al.|Wang et al.|Shvetsova et al.|Recasens et al.|Ours|
> |-|-|-|-|-|-|-|
> |RGB, Audio, OCR, Speech|90.6|89.7|90.0|90.6|90.5|**79.4**|
> |RGB, Object, Scene, Face|89.1|88.8|88.9| 89.3|89.4|**80.3**|
> |RGB, Object, Speech, OCR|92.4|90.2|91.5|78.1|77.4|**70.2**|
> |RGB, Scene, Audio, OCR|91.0|89.9|90.7|75.3|74.2|**70.3**|
> |RGB, Scene, Speech|95.6|94.5|95.1|80.2|79.3|**74.3**|
> |RGB, Object, Audio|96.1|96.0|96.3|82.1|80.3|**76.9**|
> |RGB, Speech|98.3|98.0|98.8|84.6|83.0|**81.4**|
> |RGB, Audio|98.0|97.3|98.4|85.3|84.5|**79.8**|
> Table: Multimedia retrieval
>
> |Model|Gabeur et al.|Nagrani et al.|Wang et al.|Shvetsova et al.|Recasens et al.|Ours|
> |-|-|-|-|-|-|-|
> |Image, Depth|1.40|1.39|1.41|1.38|1.41|**1.29**|
> |Force, Proprioception|1.39|1.37|1.39|1.37|1.39|**1.27**|
> |Depth, Proprioception, Force|1.48|1.45|1.44|1.37|1.34|**1.19**|
> |Image, Proprioception, Force|1.47|1.43|1.45|1.32|1.30|**1.18**|
> |Depth, Force|1.77|1.72|1.79|1.62|1.58|**1.47**|
> |Depth, Proprioception|1.60|1.52|1.55|1.49|1.43|**1.38**|
> |Image, Force|1.71|1.64|1.70|1.58|1.52|**1.44**|
> |Image, Proprioception|1.54|1.50|1.52|1.38|1.35|**1.27**|
> Table: Robotic State Regression
>
> Since Gabeur et al., Nagrani et al. and Wang et al. assume modality-complete data, they obtain worse performance with unseen combinations than with seen combinations (top two rows), even with additional modalities. Thus we conclude these methods overfit to seen combinations. Since Shvetsova et al. and Recasens et al. aim to be robust to some modality-incomplete training data, they benefit from some modality combinations. However, these methods are not generalizable to all combinations. In contrast, our method is the most effective on all seen and unseen modality combinations.
>
> **Generalizability.** The tables above also show the generalizability of our method to different sensor combinations.
>
> **Input modality order.** We do not require input modalities to be provided in a set order since we project the features of each modality into a shared space before summation (L105-107).
>
> **Literature grouping.** We will move the related work to be section 2 to group the literature discussion.
>
> **Limitations of prior work.** The limitations in [30] also exist in [21,22,40] as these works assume modality-complete data is available like [30]. We will add clarification.
>
> **Method Details.** We will add more context to Eq. 2 and add other requested details to the method. We train our method with 120 epochs on video classification with an lr of $10^{-4}$, reduced to $10^{-5}$ for the last 50 epochs. On robot state regression and multimedia retrieval, we train with 50 epochs and an lr of $10^{-2}$. We will release the code provided in the supplementary on publication to ensure reproducibility.
>
> **Alignment.** We project the features of different modalities into the same space by the alignment loss before fusing them via a sum.
>
> **Dataset splits.** We use the same validation and test splits as provided with the datasets. To facilitate our research on unseen modality interaction, we divide the original training set of each dataset into multiple splits where each split contains different modalities. We will release our training set division.
>
> **Table numbering.** We will correct table indexes.
>
> **Modality-complete data in testing.** We mean we use modality-complete data at inference in Tables 1-3. We will make this clearer.
>
> **Implementations.** We re-implement Recassens et al. since their code is not available. We use released code for all others. We will clarify this.

---

> > ### Comment · Reviewer_2qSh · 2023-08-11
> > **Still on Table 2**
> >
> > I thank the authors for the detailed responses to my concerns. They clarify a number of points raised in my first review.
> > However, I still do not fully understand Table 2 and how I should interpret it. Considering the proposed approach focuses on settings where some of the modalities at inference time might not be available at training time, I do not quite understand how I should read Table 2. In particular, is each column referring to a specific combination at inference time? If yes, what happens in the Multimodal approach? What modalities are employed at training time in such cases? Considering the way the setting is defined, I would have said only one, but then where is the multimodality?
> > Thanks in advance for your clarifications

---

> > > ### Author Response · Authors · 2023-08-12
> > > **Clarification on Table 2**
> > >
> > > ***We thank the reviewer for engagement and encouragement***.
> > >
> > > The reviewer is correct that in this ablation we only have one modality for each training sample (e.g., either RGB only or audio). However, we have two modalities (the unseen modality combination) during inference. For each column, we consider two different modalities to study the unseen modality combination at inference. While each video sample in EPIC-Kitchens contains all the three modalities, we divide the original training set into two splits and let only one modality be available in each split during training. We leave the test set as is. For example, for the RGB & Audio column, one training split has RGB available only and the other training split has audio available only. During inference, both of the two modalities are available.
> > >
> > > We train an unimodal encoder on each split for the video classification task. The performance of these unimodal encoders are reported in the three rows of the `unimodal’ part in Table 2. For the multimodal part in Table 2, the late fusion indicates we directly average the predictions from the unimodal encoders of the two modalities during inference. For the rest of the multimodal approaches, we send a single modality into each variant of our multimodal model during training while we send both modalities into the model at inference time.
> > >
> > > We will expand the setting description to make this clearer.

---

> > > > ### Comment · Reviewer_2qSh · 2023-08-18
> > > > **Thanks again for the feedback**
> > > >
> > > > I thank the authors again for their detailed response. I will increase my rating

---

### Official Review · Reviewer_adMX · 2023-07-05

**Soundness:** 2 fair
**Presentation:** 2 fair
**Contribution:** 2 fair
**Rating:** 5
**Confidence:** 4

**Summary:**

This paper introduces a method that can enhance the performance of multimodal models in scenarios involving unseen modality interactions.

**Strengths:**

1, The issue of "unseen modality interaction" explored in this paper is quite novel.

2, This paper maps the features of different modalities into the same space and merges different modalities by simple addition. As the number of modalities increases, the number of parameters required for fusion does not significantly increase.

**Weaknesses:**

1, The performance of the baseline in this paper is so low that it makes the entire method proposed by the paper unconvincing. Specifically, although the method of this paper surpasses its own set baseline methods on EPIC-Kitchens, it only achieves an accuracy of 23.7%. Meanwhile, the most naive baseline in the paper [1] also has an accuracy of 23.7%. Therefore, it's hard for me to be convinced by the experimental conclusions of this paper.

2, In Table 2, the authors compare the performance of different methods on different datasets, but why can't many of these methods outperform a simple late-fusion? Is it because there are significant differences in certain settings? If the author cannot clarify the situation here, I would consider the experimental comparison to be very unfair.

3, Also in Table 2, the performance of DEQ Fusion in MM-IMDB is 61.52/53.38, while the performance of MMBTl[2] is 66.8/61.6 (Micro F1/Macro F1). Why choose not to report this method? It's a well-known classic approach that has already one hundred of citations.

[1] What Makes Training Multi-modal Classification Networks Hard?

[2] Supervised Multimodal Bitransformers for Classifying Images and Text

**Questions:**

See Weaknesses

**Limitations:**

I don't think the experimental results of this paper convince me, so I tend to reject this paper.

---

> ### Author Rebuttal · Authors · 2023-08-08
>
> ***We thank the reviewer for their time and effort. We are glad to hear the reviewer found our proposed problem of “unseen modality interaction” novel and that the reviewer appreciates that  number of parameters required for fusion does not significantly increase as the number of modalities increases***.
>
> &nbsp;
>
> **Clarification on the experimental setups.** Our results on EPIC-Kitchens are not comparable to the results of Wang et al. [1] as we aim for unseen modality interaction where we do not have access to data with all modalities present. The A/V baseline from Table 8 in [1] instead assumes all samples have all modalities present.
>
> Specifically, we modify the training set of EPIC-Kitchens by dividing the training data into two splits with each split having a different set of modalities. The A/V baseline is trained with the full training set. Training the naive A/V from [1] with our unseen modality training splits, we get 19.0%, much lower than our result (23.7%). We will add this baseline to the final version.
>
> **Late fusion vs. recent methods.** Our late-fusion baseline outperforms recent multi-modal fusion methods as late fusion simply averages the predictions from unimodal models. Previous multi-modal fusion models learn correspondences under the assumption that some or all of the data is modality-complete. When there is no modality-complete data for model learning, these models cannot learn cross-modal correspondences and instead overfit to the modality combinations seen during training. We realize the term ‘late-fusion’ can be ambiguous and will clarify it refers to averaging the final predictions from unimodal encoders in our results.
>
> **Comparison with MMBT.** Even though this comment seems to be for another paper, we further compare our paper with MMBT on EPIC-Kitchens, and report the performance in the table below:
>
>
> | | Top-1 (%) |
> |-------|---|
> | MMBT |   17.4  |
> | This paper| 23.7 |
>
>
> While MMBT is an effective method for modality-complete multimodal fusion, we conclude that our method is more effective in unseen modality interaction.
>
> [1] Wang et al. "What makes training multi-modal classification networks hard?." In CVPR 2020.

---

> > ### Comment · Reviewer_adMX · 2023-08-11
> >
> > Thank you to the author for responding to my questions and pointing out my mistakes. I have raised my score to borderline accept.

---

### Official Review · Reviewer_ctBz · 2023-07-06

**Soundness:** 2 fair
**Presentation:** 2 fair
**Contribution:** 3 good
**Rating:** 5
**Confidence:** 4

**Summary:**

This paper tackles the issue of unseen modality interaction, which challenges the conventional assumption of modality completion during training. The approach taken in this study involves formulating a training setting that accounts for modality incompleteness. Subsequently, the proposed method focuses on projecting multiple modalities into a shared feature space through an alignment objective and leveraging a pseudo-labeling strategy to alleviate the model's tendency to overfit to unreliable modalities. The experimental results on multiple benchmarks demonstrate the efficacy of the proposed framework.

**Strengths:**

- The paper addresses a practical problem, since incompleteness of modality occurs often in reality.
- The authors conduct their experiments on the datasets with various modalities.

**Weaknesses:**

- There are some ambiguous parts in the manuscript.
    - The authors mention that L_{align} ensures the projection of features from different modality spaces into the same feature space. However, based on Eq (1), it appears that the loss is computed as the summation of the difference between modality-specific features and modality-specific learnable tokens. In light of this, how can we ensure that the features from all modalities are projected into the same feature space?
    - Regarding the dual branch prediction part, please provide further elaboration on the intuition behind the pseudo-labeling strategy to address less discriminative modalities. Can we consider less discriminative modalities as unreliable modalities? Even if a modality is less discriminative than others, it may still contain important information for the model. I am curious about the purpose of using pseudo labels to suppress such less discriminative modalities or encourage them.
- The numbering of tables is incorrect in the experimental section. For instance, the table referenced in line 160 should be Table 1, not Table 6. Most of the tables are misnumbered.
- Line 273 needs to be ended with ‘.’

**Questions:**

- In a similar context, is there any reason to split $\hat{F}$ into exactly half? Since the contribution of $L_{pseudo}$ is extremely far from $L_{supervised}$, may be dividing $\hat{F}$ into half is not the optimal solution.
- In line 146, the paper mentions selecting the modality-specific pseudo-label that is closest to the ground truth annotation. How is the distance measured between the label and ground truth annotation when using a one-hot vector label?
- Why the number of video classification accuracy in Table 2, 3 (23.7%) and Table 4 (23.8%)?

**Limitations:**

The paper sufficiently deals with the limitation of the paper in the conclusion.

---

> ### Author Rebuttal · Authors · 2023-08-08
>
> ***We thank the reviewer for their time and effort and are glad that the reviewer found our paper addresses a practical problem and appreciate the experiments conducted on various modalities***.
>
> &nbsp;
>
> **Alignment Loss.** We apologize for the unclear text. The learnable tokens are not modality-specific but are shared across modalities. We make this clearer by modifying Eq.1 as:
>
> \begin{aligned}
> L_{align} &= \sum_{m \in M_1} || \bar{f_m} - u_{n_m} ||^2_2,
> \end{aligned}
>
> where $u_{n_m}$ is the learnable token from $[u_1, …, u_{n_u}]$ selected for feature $\bar{f}_m$. With the shared learnable tokens, the modality-specific features are encouraged to be projected into a common feature space.
>
>
> **Intuition behind pseudo-labeling strategy.** We agree with the reviewer that even if the modality is less discriminative, it can still contain important information. Our intuition for the pseudo-labeling strategy is inspired by the observation that a single modality alone often cannot provide enough information for accurate prediction. Take the example of conducting activity recognition with audio and video modalities, the audio is often less discriminative than video. For instance, the audio modality can be crucial in distinguishing that the activity is one of *swimming*, *surfing* or *water skiing*, but cannot make fine-grained distinctions. By forcing the model to predict the ground-truth activity *swimming*, it may overfit to some unrelated features such as background noise. By using average predictions as pseudo-labels to provide a distribution over classes, the model is able to incorporate the important distinguishing information while avoiding such overfitting as it allows uncertainty between multiple classes. We will make this motivation clearer in the final version.
>
>
> **Typos.** We thank the reviewer for highlighting the incorrect table indexes and the typo. We will correct them.
>
> **Dividing $\hat{F}$ in half.** We add experiments to test the model’s performance with different partition strategies on EPIC-Kitchens and report the results in the table below.
>
> | Ratio of $L_{pseudo}$ to $L_{supervised}$ | Top-1 (%) |
> |-------|---|
> | 30:70 |   22.9   |
> | 50:50 | **23.7** |
> | 70:30 |    22.5    |
>
>
> Dividing the tokens into half delivers the best performance. While the pseudo-labels can refine the overconfident predictions trained by groundtruth labels, they may also be noisy for some difficult samples while the groundtruth labels provide the correct supervision. Thus, making the pseudo supervision and the groundtruth supervision equally important is beneficial.
>
>
> **Distance measurement for one-hot labels.** When using one-hot vector labels, the distance is measured by the cosine similarity between the modality-specific prediction and the groundtruth one-hot vector label. We will add this to the final version.
>
> **Number inconsistency.** We apologize for this mistake, the accuracy in Table 4 should be 23.7%.

---

> > ### Comment · Reviewer_ctBz · 2023-08-15
> > **Thanks for the response.**
> >
> > Thanks for the responses. I still want to discuss the following points.
> >
> > Regarding $L_{align}$, is it correct to say that $u_{n_{m}}$ is not uniquely determined for each modality? In other words, within the set $[u_{1}, …, u_{n_{u}}]$, is it possible for features from different modalities $(f_i, f_j)$ to be assigned to the same $u_{k}$? This scenario would occur if $u_{k}$ is the nearest token for both modalities. Consequently, does the proposed objective facilitate the projection of features from distinct modalities into a shared space? It’s the authors’ claim, right? Please kindly correct me if there are any misconceptions in my understanding.
> >
> > For the pseudo-labeling, I now understand what the authors tried to do after reading the response. Within this context, I believe that the paper could benefit from a more comprehensive exploration of the pseudo-labeling process. Specifically, the inclusion of analysis or qualitative results of the pseudo-labels generated for various modalities would support the authors' claim and enhance the overall robustness of the paper.
> >
> > One more thing to point out is that I think the sentence “the RGB modality is more discriminative than optical flow or audio” in line 124 should be revised. For video classification [A, B], optical flow often performs better than RGB, so the statement is not always correct.
> >
> > [A] : Carreira et. al, Quo vadis, action recognition? a new model and the kinetics dataset, CVPR 2017,
> >
> > [B] : Wang et. al, Temporal segment networks: Towards good practices for deep action recognition, ECCV 2016

---

> > > ### Author Response · Authors · 2023-08-15
> > > **Further Clarifications**
> > >
> > > **We thank the reviewer for engagement and the opportunity for further clarification**.
> > >
> > > **$L_{align}$**. The reviewer is correct that  $u_{n_{m}}$ is not uniquely determined for each modality. It is possible for features from different modalities $f_i, f_j$ to be assigned to the same $u_k$ when $u_k$ is the nearest token for both modalities. The proposed objective does facilitate the projection of features from distinct modalities into a shared space. We further verify this claim by computing the feature distance between modalities on EPIC-Kitchens with the variants of our model used in Table 2 of the main paper (we provide more explanations for Table 2 in the response to reviewer 2qSh). Specifically, we first obtain the average feature before the multimodal transformer per class for each modality and then compute the average Euclidean distance between modalities across classes. For RGB & Audio, after adding the feature projection to the vanilla transformer, the average Euclidean distance reduces from 84.1 to 75.4. For RGB & Flow, the distance reduces from 81.3 to 72.5 and for Audio & Flow, it drops from 83.9 to 76.2. We will add this into the paper.
> > >
> > > **Comprehensive Exploration of Pseudo-Labeling**. In the second row of Table 3, we show that when using the same supervised loss with groundtruth for both branches instead of the pseudo-labeling, our multimodal model suffers a 2.3% accuracy decrease. This is because the pseudo-labeling eliminates the overfitting to groundtruth labels, which can be harmful when a particular modality combination cannot give a reliable prediction. We also observe that when using the pseudo-labeling, the validation accuracy becomes higher than the groundtruth-supervision only, with the same number of epochs. Since we cannot provide figures in the discussion phase, we describe several examples for various modalities here and will provide the qualitative examples in the appendix.
> > >
> > > For the RGB modality, given a video sample of taking bowl, the pseudo-label has a probability of 0.60 for *take bowl* while 0.30 for *take cup* since the activity happens out of view and it is hard to judge whether there is a bowl or cup. For the audio modality, given an audio track of opening cupboard, the pseudo-label has a probability of 0.50 for *open cupboard* and 0.45 for *open fridge* as the sounds are similar. For the optical flow modality, given a video sample of taking soy milk, the pseudo-label has a probability of 0.35 for taking soy milk while 0.65 for milk since the objects have a similar motion appearance. As a result, forcing our multimodal model to be far away from any of the similar activity classes would result in overfitting. We will add the clarifications into the paper.
> > >
> > > **Inappropriate Statement**. The reviewer is quite right that our statement on RGB discriminability is far too general. We will modify the sentence “the RGB modality is more discriminative than optical flow or audio” into “for activity recognition in EPIC-Kitchen videos, the RGB modality is often more discriminative than audio”. Thank you.

---

> > > > ### Comment · Reviewer_ctBz · 2023-08-17
> > > > **Finalize rating**
> > > >
> > > > Thanks to the author for responding to my questions. Although I believe the framework is not super-effective or fancy, and there's room for enhancement in the manuscript, this submission is valuable as a pioneer of a novel and practical problem setup. Therefore, I raised my rating as borderline accept.

---

### Decision · Program_Chairs · 2023-09-21

**Decision:**

Accept (poster)

**Comment:**

The reviewers found the paper to be making a valuable contribution, to a large degree by exploring a practical issue that has received little attention so far, and recommend it for acceptance, trusting the authors to make the manuscript clarifications they promised in their rebuttals.